# Effect of Small-Molecule Natural Compounds on Pathologic Mast Cell/Basophil Activation in Allergic Diseases

**DOI:** 10.3390/cells13231994

**Published:** 2024-12-03

**Authors:** Robert Werner, Michelle Carnazza, Xiu-Min Li, Nan Yang

**Affiliations:** 1Division of R&D, General Nutraceutical Technology LLC, Elmsford, NY 10523, USA; robert.werner@gnt-us.com (R.W.); michelle.carnazza@gnt-us.com (M.C.); 2Department of Pathology, Microbiology & Immunology, New York Medical College, Valhalla, NY 10595, USA; xiumin_li@nymc.edu; 3Department of Otolaryngology, School of Medicine, New York Medical College, Valhalla, NY 10595, USA; 4Department of Dermatology, New York Medical College, Valhalla, NY 10595, USA

**Keywords:** mast cell/basophil activation, allergic diseases, pathologic IgE, small-molecule natural compounds

## Abstract

Pathologic mast cells and basophils, key effector cells in allergic reactions, play pivotal roles in initiating and perpetuating IgE-mediated allergic responses. Conventional therapies for allergies have limitations, prompting exploration into alternative approaches such as small-molecule natural compounds derived from botanical sources. This review synthesizes the existing literature on the effects of these compounds on pathologic mast cells and basophils, highlighting their potential in allergy management, and utilizes the PubMed database for literature acquisition, employing keyword-based searches to identify relevant peer-reviewed sources. Additionally, mechanistic insights were evaluated to contextualize how small-molecule natural compounds can inhibit mast cell/basophil activation, degranulation, and signaling pathways crucial for IgE-mediated allergic reactions. Small-molecule natural compounds exhibit promising anti-allergic effects, yet despite these findings, challenges persist in the development and translation of natural compound-based therapies, including bioavailability and standardization issues. Future research directions include optimizing dosing regimens, exploring synergistic effects with existing therapies, and employing systems pharmacology approaches for a holistic understanding of their mechanisms of action. By harnessing the therapeutic potential of small-molecule natural compounds, effective treatments for allergic diseases may be realized, offering hope for individuals with allergies.

## 1. Introduction

Allergies, which can be life-long and lead to severe reactions, currently lack a cure. Existing treatments, such as allergen avoidance, monoclonal antibodies, or oral immunotherapy (OIT), have limitations and are not always practical [1,2,3]. Omalizumab, an anti-IgE antibody, can bind free IgE, preventing its interaction with the high-affinity IgE receptor (FcεRI) on mast cells and basophils, but it does not affect IgE production by long-lived plasma cells or IgE already bound to FcεRI [4]. OIT, including treatment with Palforzia, a prescription medication used to treat peanut allergy, must be administered daily to maintain the treatment effect; it is only intended to decrease sensitivity to small amounts of peanuts, and it is not effective against other food allergies [5]. Thus, there is a pressing need for safer and more effective long-term treatments for IgE-mediated allergic conditions.

Central to the initiation and progression of allergic reactions are mast cells and basophils [6], two key effector cells of the immune system. In many ways, they are sibling cell types due to their overlapping immune functions as mediators of immunoglobulin E (IgE)-dependent degranulation [7], in which they share many of the same granule components [8,9]. Representing minor cell populations, basophils and mast cells arise from CD34+ hematopoietic stem cells and constitute less than 1% of peripheral blood and bone marrow cells [10]. Basophils primarily stay within the bone marrow, where they mature before entering circulation for a brief lifespan, whereas mast cells leave the bone marrow as immature precursors and undergo maturation within the tissue, where they establish long-term residency [11,12,13].

Although they are a small fraction of the overall immune cell population, mast cells and basophils play pivotal roles as effector cells in type I hypersensitivity reactions, such as allergy. During the sensitization phase of an allergic response, allergens are taken up by antigen-presenting cells (APCs) like dendritic cells, processed, and presented to T-helper (Th) cells. Once the allergen is recognized as foreign, Th cells become activated, stimulating B cells to produce allergen-specific IgE antibodies. These IgE antibodies subsequently bind to FcεRI on the surface of mast cells and basophils. Upon re-exposure to allergens, mast cells and basophils become activated, leading to the release of presynthesized inflammatory mediators, including histamine, heparin, various proteases, and proteoglycans, which contribute to the characteristic symptoms of allergies, such as itching, swelling (angioedema), respiratory distress (coughing, wheezing, and shortness of breath), gastrointestinal symptoms, and fatal or near-fatal anaphylaxis. Additionally, following degranulation, the de novo synthesis of cytokines, chemokines, interferons, cysteinyl leukotrienes, prostaglandins, and several growth factors play immunomodulatory roles, including the initiation and magnification of inflammation, as well as leukocyte migration, proliferation, and activation [14,15,16].

There is growing interest in exploring alternative approaches, particularly the use of small-molecule natural compounds isolated from botanicals, to modulate the activity of pathologic mast cells and basophils. These bioactive small-molecule natural compounds are derived from plants, fungi, and other natural sources, and are known for their diverse pharmacological properties and relatively low toxicity [17,18]. Research into the effects of small-molecule natural compounds on pathologic mast cells and basophils focuses on their role in inhibiting activation and degranulation, modulating mediator release, and regulating cell signaling pathways [19].

The modulation of mast cell and basophil activity by small-molecule natural compounds represents a promising area of research that could lead to the development of more targeted approaches to allergy treatment [20]. Understanding the mechanisms by which these compounds exert their effects on pathologic mast cells and basophils can advance our knowledge of allergic disease pathophysiology. Therefore, this review aims to synthesize and evaluate the existing literature on the subject, offering insights into the mechanisms of action of specific small-molecule natural compounds and their potential applications in allergy management.

## 2. Mast Cell and Basophil Function in Allergy

Mast cells and basophils are pivotal players in IgE-mediated allergic diseases such as asthma, rhinitis, food allergy, and atopic dermatitis because they express the high-affinity IgE cell surface receptor, FcεRI. The signaling pathways that regulate FcεRI-mediated mast cell and basophil activation have been thoroughly investigated. In brief, the FcεRI receptor is a heterotetrametric protein composed of one alpha chain (FcεRIα), one beta chain (FcεRIβ), and two disulfide-bonded gamma chains (FcεRIγ) [21]. The FcεRI dependent degranulation process begins when IgE bound to FcεRI recognizes an antigen. The IgE-FcεRI complex then crosslinks to nearby IgE-FcεRI complexes and that triggers a subsequent phosphorylation cascade orchestrated by the src-family tyrosine kinases, particularly Fyn and Lck/yes-related novel tyrosine kinase (Lyn), which are non-receptor protein kinases that phosphorylate tyrosine residues found within the immunoreceptor tyrosine-based motif (ITAM) on FcεRIβ and FcεRIγ [22,23]. Lyn is conserved across species, broadly expressed, and propagates both inhibitory and activating pathways in the immune cell lineage [24]. Lyn or Fyn kinase activity enables the recruitment of Spleen tyrosine kinase (Syk) to FcεRIγ through Syk’s Src-homology 2 (SH2) domain, a roughly 100 amino acid long sequence that binds to phosphorylated tyrosine residues and allows Syk to interact with FcεRIγ [25]. Activated Syk further phosphorylates downstream targets including Linker for activation of T-cell (LAT) protein, which assembles into a complex signaling hub [26]. The signaling cascade diverges into multiple routes, including the downstream activation of phospholipase Cγ (PLCγ), phosphatidylinositol-3-kinase (PI3K), protein kinase B (Akt), inositol triphosphate (IP3), diacylglycerol (DAG), protein kinase C (PKC), and the MAP kinases (ERK, JNK, and p38). This leads to the activation of transcription factors involved in cytokine generation and arachidonic acid metabolism, such as the generation of prostaglandins and leukotrienes [27]. Furthermore, it triggers calcium mobilization, which is crucial for mast cell/basophil degranulation and pathogenicity if left unregulated [28].

Counterbalancing these positive signaling pathways are phosphatases like Src-homology 3 (SH3) domain-containing protein tyrosine phosphatase 1 (SHP-1) and SH2-containing inositol polyphosphate 5 phosphatase (SHIP). Negative regulators like SHP-1 and SHIP play crucial roles in dampening mast cell and basophil activation. They help maintain homeostasis, allowing these cell types to return to their basal, non-activated state [29]. In addition, liver kinase B1 (LKB1) and AMP-activated protein kinase (AMPK) have been identified as additional negative regulators for FcεRI signaling in mouse mast cells [30], where LKB1 is an upstream kinase of AMPK (Figure 1). This suggests that activators of SHP-1, SHIP, and AMPK could be promising therapeutic candidates for allergic diseases [31].

## 3. Review of Botanical Drugs and Small-Molecule Natural Compounds on Mast Cells and Basophils

Small-molecule natural compounds have gained much attention for their potential to modulate immune responses, immunometabolism, and inflammatory pathways [32]. In the context of mast cells and basophils, investigations into the effects of small-molecule natural compounds have focused on their ability to inhibit pathologic activation, degranulation, and inflammatory mediator release, as well as to regulate cell signaling pathways involved in allergic reactions. These compounds offer promising avenues for the development of novel therapeutic agents in allergy management. Still, several gaps exist in the field of small-molecule natural compounds and their effects on pathological mast cells and basophils. This includes the limited understanding of their mechanisms, interaction with other cell types and systems, centralized documentation of small-molecule natural compounds and their effects on mast cells and basophils, and the clinical standardization of medications so that they can be better translated to treatments. Based on these criteria, several nutraceuticals have shown potential in stabilizing mast cells and basophils to control allergic reactions (Table 1).

### 3.1. Berberine

Berberine, an isoquinoline alkaloid found in medicinal herbs like *Coptis chinensis*, *Phellodendron japonicum*, and *Berberis aquifolium*, exhibits diverse biological activities such as anti-diabetic, anti-cancer, and anti-inflammatory properties. It is widely used in China for managing glycemic levels in type 2 diabetes. Berberine’s clinical effectiveness in diabetes is attributed to its ability to activate AMPK, like metformin, which itself is derived from galegine, a natural compound from the plant *Galega officinalis* [55,56,57]. Of the compounds isolated from *Phellodendron*, it was reported that berberine showed the most potent inhibitory effect on mast cell degranulation than other isolated compounds, including palmatine and jatrorrhizine [33] (Figure 2). Mechanistically, studies utilizing Western blot analysis on RBL-2H3 cells revealed that berberine can hinder mast cell degranulation by suppressing Lyn, Syk, and Gab2 phosphorylation compared to control cells. When DNP-IgE primed RBL-2H3 cells were pretreated with 0.3, 3, or 30 µM of berberine and subsequently stimulated with DNP-HSA, they exhibited a dose-dependent reduction in β-hexosaminidase release and histamine levels. MAPK signaling was also attenuated as measured by a decrease in phosphorylated JNK, ERK, and p38. Subsequently, the decrease in MAPK signaling led to a decrease in IL-4 and TNF-α production, which can suppress external immune cell recruitment and activation [18,34,35]. Although berberine’s activation of AMPK has not been directly measured in Ag/IgE/FcεRI signaling, berberine treatment both in vivo and in vitro is able to attenuate basophil and mast cell degranulation and significantly lower phosphorylated Syk [33]. Future studies should address the role of AMPK in FcεRI signaling to confirm if AMPK acts as a negative regulator to mast cell and basophil degranulation. Additionally, more studies need to be performed to determine the extent to which berberine activates AMPK in mast cells and basophils and to explore if berberine exerts an immunomodulatory effect through other mechanisms.

The therapeutic effects of berberine have also been examined to treat atopy-like symptoms in mice. In an atopic dermatitis model where NC/Nga mice were sensitized to mite allergens (Der p), berberine administration significantly reduced the clinical skin severity score, spontaneous scratching behavior, and serum IgE levels in the mice. This effect was not observed in untreated mice, suggesting a targeted action of berberine in pathological conditions. Furthermore, berberine significantly decreased the number of cutaneous eosinophils and mast cells in a dose-dependent manner and attenuated the number of degranulated mast cells by histological analysis using H&E and toluidine blue staining. On a molecular level, berberine lowered the mRNA and protein levels of eotaxin (CCL11), a chemokine that attracts eosinophils, as well as decreasing the mRNA and protein levels of macrophage migration inhibitory factor (MIF) and the Th2 cytokines IL-4 and IL-5 in the skin of dermatitis mice, while having no effect on the Th1 cytokine and IFN-γ. Additionally, in MC/9 murine mast cells sensitized with anti-DNP IgE, berberine suppressed the expression of MIF and IL-4 and inhibited histamine release. The reduction in IL-4 in mast cells is notable because IL-4 is crucial for class switching in B cells to IgE-secreting plasma cells [36]. Consequently, this downregulation of IL-4 could lead to a feedback loop, resulting in decreased IgE production and decreased binding of free IgE to FcεRI. Gene expression analysis using a GeneChip microarray revealed that berberine downregulated several inflammation-related genes, including ELF3F and MALT1, which were further confirmed at the protein level through Western blotting and siRNA knockdown of ELF3F and MALT1 [37]. ELF2F and MALT1 knockdown led to a decrease in the expression of IL-4 and MIF, providing evidence that suggests berberine attenuates ELF2F and MALT1, which then reduces the expression of IL-4 and MIF. Given the widespread use of berberine in traditional Chinese medicine, these findings provide a scientific basis for its continued and expanded use in modern medical practices [58].

One challenge of using berberine in clinical practice is its poor oral bioavailability [59,60,61]. This limitation hinders its therapeutic effectiveness, as insufficient levels of berberine reach systemic circulation to exert the desired biological effects [62]. To address this issue, different strategies have been investigated to improve berberine absorption and bioavailability, including modifications in herbal extraction methods and the incorporation of adjuvant herbal ingredients. Yang et al. [61] demonstrated that Caco-2 human intestinal epithelial cells can be used to test the absorption of berberine when coincubated with specific herbal constituents. Their study showed that when 50 µg/mL of berberine was coincubated with *Angelica sinensis* at a concentration of 250 µg/mL, the absorbed levels of berberine increased from 4.06 µg/mL to 12.39 µg/mL, resulting in an absorption improvement of 205.4%. Furthermore, other herbs, such as *Fructus prunimume* and *Zanthoxylum bungeanum*, also contributed to an increase in berberine absorption, achieving absorption improvements of 132.8% and 126.8%, respectively. These findings underscore the potential of using adjuvant herbal ingredients to improve the bioavailability of berberine, thereby enhancing its therapeutic effectiveness in clinical practice.

In addition to herbal adjuvants, the formulation of berberine as nanoparticles offers an additional approach to enhance berberine bioavailability. The gastrointestinal (GI) tract plays an important role in drug absorption, yet barriers such as mucus, epithelial tight junctions, efflux transporters, and metabolic enzymes can degrade berberine before it reaches its intended target. By encasing berberine in nanoparticles, the challenges of berberine’s poor bioavailability can be mitigated, allowing for a prolonged release that extends berberine’s presence in the GI tract while minimizing rapid metabolism and degradation. For example, berberine-containing chitosan nanoparticles achieve a controlled release, with approximately 48.14% released within 12 h and 74.30% over 48 h, compared to a rapid release from free berberine solutions [63]. This prolonged release is attributed to the protective chitosan matrix surrounding the berberine, improving its absorption and therapeutic efficacy. Overall, the use of nanoparticles not only enhances berberine’s bioavailability but also highlights the innovative potential of nanotechnology in optimizing drug formulations for better clinical outcomes.

### 3.2. Sauchinone

Sauchinone, a biologically active lignin derived from *Saururus chinensis*, known for its anti-cancer and anti-inflammatory properties, has been evaluated for its ability to mitigate allergic reactions in a mouse model of anaphylaxis [38,39]. Mice were sensitized with anti-DNP-IgE and challenged with DNP-HSA and showed decreased serum levels of Leukotriene C_4_ (LTC_4_) and prostaglandin D_2_ (PGD_2_) after oral administration of sauchinone, suggesting that sauchinone attenuates phospholipase signaling. The inhibitory effects were dose-dependent, with significant reductions in Evans blue dye extravasation at 25 and 50 mg/mL doses of sauchinone. In fact, sauchinone’s efficacy at a concentration of 50 mg/mL in suppressing FcεRI-mediated anaphylaxis was comparable to a similar dose of fexofenadine–HCL (Allegra), a widely used antihistamine. Additional in vitro studies utilizing bone marrow-derived mast cells (BMMCs) confirmed sauchinone’s efficacy in attenuating IgE/Ag-induced FcεRI signaling through the activation of the LKB1-AMPK pathway, a negative regulator of FcεRI [38]. Treatment with sauchinone led to the dose-dependent phosphorylation of LKB1, AMPK, and acetyl-CoA carboxylase (ACC), which is downstream of AMPK. A 20 µM dose of sauchinone showed optimal effects and the increase in phosphorylation was time-dependent, peaking at 120 min. Sauchinone also inhibited the activation of downstream signaling molecules such as PLCγ1, ERK1/2, JNK, and IKK.

Independent of the LKB1/AMPK axis, sauchinone induced SHP-1 phosphorylation, a negative regulator of Syk. Silencing SHP-1 using siRNA-treated BMMCs diminished sauchinone’s suppressive effects on Syk phosphorylation and led to increased mast cell degranulation. This dual mechanism of action elucidates the anti-allergic activity of sauchinone and highlights its potential as a therapeutic agent for allergic inflammatory diseases by increasing phosphorylated SHP-1, LKB1, AMPK, and ACC [39]. Additionally, comparative studies using sauchinone and other standard allergy treatments like fexofenadine could provide insights into its relative efficacy and potential advantages or disadvantages compared to established treatment regimens.

### 3.3. Arctigenin

The overproduction of IgE drives the pathology of mast cells and basophils, emphasizing the need for therapeutic interventions to modulate this pathway. Recent investigations have shown that Arctium lappa, a perennial herb from the Asteraceae family, acts as a potent modulator of inflammatory pathways, including those involved in the pathogenesis of human inflammatory bowel disease [64,65]. Researchers screened approximately 300 medicinal herbs and found Arctium lappa to be one of the most effective at suppressing IgE production and Th2 cytokines (IL-5, IL-13). They isolated the two main compounds, arctiin (IC50 = 14.94 μg/mL) and arctigenin (IC50 = 5.09 μg/mL), through liquid–liquid extraction and column chromatography, with arctigenin being the more potent. In vitro studies showed that arctigenin dose-dependently inhibited IgE synthesis in IgE-producing human myeloma U266 cells, and in vivo murine models supported these findings by exhibiting significant reductions in allergen-specific IgE levels. These effects also correlated with improvements in allergy symptoms, such as a reduction in plasma histamine levels after an allergen challenge, as measured by ELISA (Figure 3). Safety assessments, including feeding mice a 15-fold daily dose of arctigenin confirmed its benign safety profile [40]. This study demonstrates that arctigenin can reduce circulating IgE levels by targeting IgE synthesis, which may mitigate the pathogenicity of mast cells and basophils by decreasing secreted IgE antibodies, particularly those reacting to innocuous antigens. These findings are similar to the outcomes seen in patients treated with omalizumab; however, arctigenin targets IgE synthesis, whereas omalizumab binds to already secreted IgE.

Kee et al. [41] also assessed the potential of arctigenin in mitigating allergic responses mediated by mast cells, specifically focusing on histamine release, inflammatory cytokine production, and signaling pathways involved in allergic reactions. Experiments on mast cell lines, HMC-1 and RBL-2H3, revealed that arctigenin dose-dependently suppressed histamine release after mast cell activation. Furthermore, arctigenin downregulated the expression of proinflammatory cytokines (IL-1β, IL-6, IL-8, and TNF-α) and chemokines (CCL2, CCL3, CCL4, and CCL5) in mast cells, indicating its potential to alleviate allergic inflammation. Further investigations into the underlying mechanisms of arctigenin on mast cells showed that it exerts anti-inflammatory effects by inhibiting the phosphorylation of MAPKs (ERK, JNK, and p38) and decreasing the nuclear translocation of NF-κB in phorbol 12-myristate 13-acetate plus calcium ionophore A23187 (PMACI)-activated mast cells. Additionally, arctigenin inhibited RIP2 and caspase-1 activation, further attenuating the inflammatory cascade initiated by allergen stimulation [41]. In a similar vein, researchers investigated the regulatory effects of Arctium lappa extract, a natural source of arctigenin, on the activation of the FcεRI cascade in mast cells. IgE-sensitized RBL-2H3 cells were treated with fermented *Arctium lappa* extract, which inhibited the phosphorylation of Lyn, Fyn, and Syk and significantly reduced the phosphorylation of the downstream targets of Syk (ERK, JNK, p38, and Akt) as well as PLCγ1/2 and PKCδ [42]. These findings underscore the promising therapeutic potential of arctigenin in mitigating pathologic mast cells and by extension basophils by targeting FcεRI signaling.

The identification of arctigenin as a modulator of cytokine release in mast cells and basophils can play an important role in the maintenance of chronic inflammation and returning the body to homeostasis. If arctigenin treatment successfully mitigates long-term proinflammatory cytokine and chemokine release, then it can be used as a new targeted treatment for allergic and inflammatory diseases by returning proinflammatory molecules to basal levels instead of suppressing them completely, like with monoclonal antibodies. Future investigations should prioritize exploring the direct impact of arctiin or arctigenin on mast cells and basophils while utilizing in vivo time course studies to measure attenuated proinflammatory cytokine release and immune cell recruitment. Additional studies can compare omalizumab treatment and arctigenin to elucidate the potential benefits and mechanisms by which arctigenin modulates immune responses, ultimately providing a more comprehensive understanding of its therapeutic potential in managing allergic and inflammatory conditions.

### 3.4. Sophoraflavanone G

Sophoraflavanone G (SFG), an isopentane diflavone derived from the widely used traditional Chinese medicine plant *Sophora flavescens*, has been identified to act as a potent Syk inhibitor using a combination of virtual screening, molecular docking, and biological assays. In the initial screening of 343 TCM compounds, SFG showed efficacy in inhibiting Syk activity compared to other molecules tested in an ADP-Glo kinase assay. Molecular docking and dynamics simulations further provided evidence for the stability of SFG binding to Syk with a predicted binding free energy of −49.27 kcal/mol. In the same experiment, other predicted molecules with potential Syk inhibitory activity include arctiin, salvianolic acid A–C, rutin, naringin, and lithospermic acid. Additionally, in vivo and in vitro studies showed that SFG significantly inhibited mast cell degranulation in IgE/BSA-stimulated RBL-2H3 cells by reducing β-hexosaminidase release and the levels of proinflammatory cytokines TNF-α and IL-4. Downstream of Syk, SFG suppressed the phosphorylation of PLC-γ1, AKT, p38, ERK1/2, and JNK. In vivo, SFG attenuated ear swelling in an IgE-mediated passive cutaneous anaphylaxis model in mice, as well as decreasing the expression of COX-2 and IL-4 as measured by ELISA, underscoring its potential as a compound for anti-allergic therapeutics [20]. Given that *Sophora flavescens* is a commonly used medicinal herb, isolating the most efficacious single molecules can provide new avenues for treatment and lower the dosing burden of plant-based formulations so that the treatment can be more widely adopted outside of a traditional medicine setting.

### 3.5. Kaempferol

Kaempferol (KAE), a naturally occurring flavonoid, shows potential in mitigating IgE-mediated anaphylaxis in C57BL/6 mice. In an IgE-mediated PCA model, KAE administration resulted in decreased edema as measured through paw thickness and dye extravasation by Evans blue administration. Quantitatively, paw thickness and Evans blue dye exudation decreased by 63.27% and 46.50%, respectively, when compared to control mice. KAE also reduced the extent of mast cell degranulation by histological analysis.

In a systemic anaphylaxis model induced by IgE/ovalbumin, pretreatment with KAE significantly reversed the drop in body temperature associated with anaphylaxis, as well as significantly lowering serum histamine levels. In vitro experiments utilizing IgE-induced activation in the human mast cell line, LAD2 cells provided further evidence for kaempferol inhibitory effect on FcεRI signaling. KAE reduced β-hexosaminidase release and histamine levels in a dose-dependent manner when LAD2 cells were pretreated with kaempferol. KAE also lowered the protein levels of TNF-α, IL-8, and monocyte chemoattractant protein-1 (MCP-1/CCL2) as measured by ELISA. Additionally, KAE was able to inhibit intracellular calcium influx, a crucial step for mast cell degranulation.

Further analysis of the upstream signaling proteins in FcεRI-mediated signaling showed a notable downregulation in the phosphorylation of PLCγ, IP3R, and PKC, indicating that KAE may exert its effect early on in the signaling cascade. KAE pretreatment also decreased the phosphorylation of Akt and NF-κB, signaling molecules essential for proinflammatory cytokine production. The predicted molecular mechanisms of kaempferol’s impact on mast cell signaling involve its inhibition of Lyn and Syk phosphorylation, key proximal kinases in the FcεRI signaling cascade, as measured through Western blot analysis in IgE-sensitized LAD2 cells. Western blot analysis also showed a dose-dependent reduction in DJ-1 levels [43]. DJ-1 has been shown to interact with Lyn and their interaction is essential for the IgE-mediated degranulation of mast cells [66]. Research by Nagata et al. [44] involved treating bone marrow-derived mast cells with KAE, which resulted in the significant inhibition of IgE-induced degranulation and cytokine production. In this set of experiments, kaempferol reduced the surface expression of FcεRI on BMMCs, though the mRNA levels of the FcεRI subunits remained unchanged. Mechanistically, KAE treatment led to increased mRNA and protein levels of SHIP1, a phosphatase that negatively regulates IgE-mediated signaling pathways in both BMMCs and peritoneal mast cells. Future research should focus on the direct impact of KAE treatment in mast cells and basophils in human models to better understand its therapeutic potential.

### 3.6. Luteolin

The flavonoid luteolin has gained attention for its potential to modulate FcεRI activation. Additionally, in mast cells, luteolin has been shown to have an inhibitory effect on FcεRI-mediated mast cell degranulation and MAS-related G protein-coupled receptor-X2 (MRGPRX2)-mediated mast cell activation also termed a pseudo-allergic reaction. Hao et al. [45] investigated the effects of luteolin on both FcεRI-mediated and MRGPRX2-mediated mast cell degranulation using both in vitro and in vivo models. Their findings revealed that luteolin had minimal cytotoxicity on mast cells and did not induce cell activation or intracellular calcium changes. However, pretreatment with luteolin effectively inhibited the release of histamine, β-hexosaminidase, 5-hydroxytryptamine (5-HT), and tryptase release in both anti-DNP-IgE-triggered mast cell degranulation and compound 48/80 (C48/80)-triggered mast cell degranulation, which is an agonist to MRGPRX2. In vivo mouse models of IgE-induced Type I hypersensitivity and C48/80-induced pseudo-allergic reactions showed that luteolin administration reduced paw swelling; Evans blue exudation; and serum levels of histamine, tryptase, TNF-α, MCP-1, PGD2, IL-8, and IL-13. Mechanistically, luteolin inhibited the increase in intracellular calcium levels and downregulated the activation of signaling molecules associated with mast cell activation pathways such as Lyn, Bruton’s tyrosine kinase (Btk), and PLC-γ.

Luteolin also decreased IgE-mediated histamine release and, to a lesser extent, calcium ionophore A23187-induced histamine in human cultured mast cells [46]. Additionally, anti-IgE-induced release of leukotrienes (LTs), PGD_2_, and GM-CSF was also inhibited by luteolin, with the strongest inhibitory effect observed in LTs. While luteolin inhibited PKC, ERK, and JNK activation, it did not show the same effect on the p38 MAPK pathway. The significance of this selective inhibition needs to be further explored to understand how it impacts the overall inflammatory response and whether there are implications for targeting specific pathways and not others.

### 3.7. Herbal Extract Formulations

#### 3.7.1. Shuang-Huang-Lian

A recent study delved into the potential anti-allergic effects of Shuang-Huang-Lian (SHL), a traditional Chinese herbal formula that has been clinically utilized to treat type I hypersensitivity. The study aimed to elucidate SHL’s molecular mechanisms of action on mast cell stabilization. Gao et al. [47] demonstrated that SHL exhibits significant efficacy in dampening mast cell degranulation induced by shrimp tropomyosin (ST) both in vitro and in vivo. SHL suppressed IgE-mediated mast cell activation, as evidenced by reduced β-hexosaminidase release in sensitized RBL-2H3 cells. In vivo studies further confirmed SHL’s protective effects against ST-induced systemic anaphylaxis, demonstrating a significant attenuation in hypothermia. One of the primary mechanisms underlying SHL’s anti-allergic effects involves stabilizing mast cells by reducing cytosolic calcium levels in resting cells. SHL achieved this by enhancing mitochondrial calcium uptake through the activation of mitochondrial calcium uniporter (MCU). The role of MCU in SHL’s action was confirmed by the loss of SHL’s effects on Ca^2+^ uptake and mast cell degranulation when MCU was genetically silenced in mice and through an MCU inhibitor in both human (LAD2) and mouse (P815) mast cells.

The study identified potential major active constituents of SHL responsible for its effects on calcium levels. Among the extracts tested, *Lonicerae Japonicae Flos* and *Fructus Forsythiae* (ELF) significantly reduced cytosolic Ca^2+^ levels, suggesting they might be the primary active constituents of SHL. Further analysis of 26 compounds from ELF identified quercetin, caffeic acid, ursolic acid, D-(−)-quinic acid, and methyl salicylate as potential small-molecule natural compounds of interest. Further studies can utilize high-performance liquid chromatography and mass spectrometry to isolate the small-molecule natural compounds of interest to test their efficacy as individual molecules or in formulations to see if small-molecule natural compounds have synergistic effects.

#### 3.7.2. FAHF-2, BF-2, and EBF-2

Several studies have investigated the efficacy of small-molecule natural compounds in preclinical and clinical settings [48,49] for the treatment of food allergies. For instance, the food allergy herbal formula-2 (FAHF-2), derived from traditional Chinese medicine, has shown promising results in inhibiting mast cell and basophil activation, thereby reducing allergic symptoms. Active compounds within FAHF-2, such as berberine, palmatine, and jatrorrhizine, have demonstrated significant inhibitory effects on IgE production and mast cell signaling pathways, including the decrease in phosphorylated Syk, highlighting their therapeutic potential in allergy management. Using a murine model of chronic peanut allergy, it was found that FAHF-2 administration prevented anaphylactic reactions following peanut challenges, with treated mice showing no symptoms and maintaining normal body temperatures, as well as reduced histamine levels compared to sham-treated mice. Additionally, FAHF-2 treatment led to a significant reduction in the number of peripheral blood basophils and peritoneal mast cells. Basophil numbers began decreasing after one week of treatment and remained lower after four weeks post-treatment. Moreover, FAHF-2 treatment suppressed IgE-mediated mast cell proliferation and degranulation in vitro, along with downregulating the expression of FcεRI on mast cells [33].

Furthermore, the combination of a modified butanol extraction of FAHF-2 with OIT was investigated because of the safety concerns surrounding the use of OIT [67,68,69]. In studies involving mice sensitized to peanuts, cashews, and walnuts, pretreatment with BF2 significantly mitigated adverse reactions [50]. Only 12% of mice receiving the BF2 + OIT treatment exhibited mild reactions, in contrast to 43% of the OIT-only group, which experienced moderate-to-severe symptoms. Additionally, the BF2 + OIT combination treatment resulted in reduced plasma histamine levels following allergen challenges. BF2 + OIT treatment also led to increased production of immunomodulatory cytokines such as IFN-γ and IL-10, alongside reductions in IL-4 and TNF-α levels, as measured by ELISA. The treatment protocols established in this study involved three groups of peanut and tree nut-allergic mice: one receiving sham treatment, another receiving OIT alone, and a third receiving BF2 (12 mg/day dissolved in drinking water) for three weeks before and during OIT. Post-treatment evaluations indicated that the BF2 + OIT regimen not only enhanced safety but also induced epigenetic modifications associated with immune tolerance, particularly in the DNA methylation of key gene promoters, including IL-4 and IFN-γ. As a result, this combined therapy may serve as a promising avenue for developing improved treatment protocols for multiple food allergies, particularly concurrent peanut and tree nut allergies. Further exploration of the synergistic effects of BF2 and OIT, especially in conjunction with established therapies like monoclonal antibodies, could provide deeper insights into their therapeutic potential and advance strategies for more effective food allergy management.

The FAHF-2 formula underwent further refinement using different solvents to enhance the concentration of berberine, resulting in the formulation termed EBF-2 [51]. In an in vivo mouse model of peanut allergy, EBF-2 treatment effectively suppressed peanut anaphylaxis by reducing peanut-specific IgE levels without affecting IgG1 or IgG2a production. EBF-2-treated mice were protected from anaphylaxis and showed lower plasma histamine levels compared to control mice. The study also revealed that EBF-2 provided long-term protection against peanut anaphylaxis and reduced the number of IgE-producing plasma cells in the spleen. This study showed that berberine inhibited IgE production and downregulated key transcription factors involved in plasma cell activation, including XBP1, BLIMP1, and STAT6. Berberine also inhibited mitochondrial respiration in IgE-producing plasma cells, potentially reducing antibody glycosylation and subsequent IgE production.

Berberine synergistically decreases FcεRI signaling by reducing phosphorylated Syk and lowers IgE generation from plasma cells, providing an interesting direction for further investigation into its potential as a therapeutic agent for allergic diseases. Both FAHF-2 and EBF-2 are Food and Drug Administration (FDA)-approved investigational new drugs. Sixty-eight subjects with a median age of 16 years old were randomized to assess the efficacy and safety of FAHF-2 [48]. One challenge to the study was adherence, with at least 44% of the subjects having poor adherence for at least one-third of the study period. The adherence was poor primarily due to the high tablet load of the medication. Participants were required to take 10 tablets three times a day for 6 months, which ended up being hard for patients 12 years old and above. Therefore, it would be beneficial to identify the active molecules in the formula and concentrate the compounds into easier-to-manage pills or test other combination therapies with established treatments, such as monoclonal antibodies or immunotherapy.

#### 3.7.3. Xin-Yi-Qing-Fei-Tang

Xin-Yi-Qing-Fei-Tang (XYQFT), composed of ten herbs, was investigated for its therapeutic efficacy using a chronic asthma mouse model and RBL-2H3 cells. In the study, BALB/c mice were intratracheally stimulated with 2.5 µg/µL of European house dust mite (*Dermatophagoides pteronyssinus*) allergen (Der p) once a week for six weeks and orally administered XYQFT at 1 g/kg 30 min before each stimulation. The study assessed airway hypersensitivity, inflammatory cell counts in the bronchoalveolar lavage fluid (BALF), and total IgE levels in the blood. Furthermore, the effects of XYQFT on mast cells’ gene expression and degranulation were evaluated using DNP-IgE-stimulated RBL-2H3 cells pretreated with XYQFT at 125, 250, and 500 µg/mL. The results demonstrated that XYQFT significantly reduced Der p-induced airway hypersensitivity, as evidenced by improved respiratory system resistance (Rrs) and elasticity (Ers). ELISA indicated a decrease in total serum IgE levels, suggesting a reduction in the Th2-mediated immune response, and the analysis of the BALF showed a reduction in inflammatory cell infiltration, including macrophages and eosinophils, following XYQFT treatment. Additionally, XYQFT inhibited the degranulation of RBL-2H3 cells and downregulated key inflammatory genes, including IL-3, IL-4, IL-13, TNF-α, GM-CSF, COX-2, ALOX-5, and MCP-1 (CCL2). LC-MS/MS was used to identify 18 compounds within XYQFT, with timosaponin AIII and genkwanin as compounds of interest for their roles in inhibiting GM-CSF and COX-2 gene expression in mast cells. Other notable small molecules are crocetin, dioscin, eudesmin, geniposidic acid, isoliquiritin, licochalcone a, liquiritigenin, liquiritin, luteoloside, neomangiferin, pseudoprotodioscin, schaftoside, eriodictyol, iridin, magnolin, and scutellarein. The proposed mechanisms include the inhibition of mast cell degranulation, reduction in inflammatory mediator release, and suppression of inflammatory cell maturation, contributing to the observed reduction in airway hypersensitivity, inflammatory cell infiltration, and elevated serum IgE levels [52].

#### 3.7.4. Jiu-Wei-Yong-An

The therapeutic potential of Jiu-Wei-Yong-An (JWYA) was investigated for its role in mitigating symptoms associated with atopic dermatitis (AD). The researchers sensitized mice with 2,4-dinitrochlorobenzene (DNCB) to mimic AD. Their study evaluated dermatitis score, skin thickness, scratching behavior, relative skin moisture content, histopathological changes, serum IgE levels, and inflammatory cytokine production, and tried to elucidate the molecular pathways involved. Notably, JWYA exhibited robust anti-inflammatory effects comparable to dexamethasone, attenuating dermatitis severity, reducing epidermal thickness, and inhibiting mast cell aggregation. Moreover, JWYA demonstrated significant efficacy in alleviating scratching behavior and restoring skin moisture content, indicative of its potential to ameliorate pruritus. Mechanistically, JWYA exerted its therapeutic effects by modulating key signaling pathways, including JAK1/STAT3 as well as p38, ERK, and JNK, which play pivotal roles in orchestrating inflammatory responses in AD. The authors identified 13 main active compounds, such as arctigenin, kaempferol, luteolin, and forsythin, as exhibiting high binding affinity for JAK1, suggesting their potential as key mediators of JWYA’s anti-inflammatory action. Furthermore, JWYA effectively suppressed the production of proinflammatory cytokines, including TNF-α, IL-1β, IL-4, IL-13, IL-31, IL-33, and IFN-γ, thereby restoring immune homeostasis in the AD murine model [53].

#### 3.7.5. *Viola yedoensis* Makino Anti-Inching Compound

*Viola yedoensis* Makino anti-inching compound (VYAC), a traditional herbal formula containing three herbs, namely *Viola yedoensis*, *Sophora flavescens* root, and *Dictamnus dasycarpus* root and rhizome, revealed significant therapeutic effects in ameliorating symptoms of DNCB-induced AD in mice. The chemical profiling of VYAC using LC-TOF-MS/MS identified 120 major constituents, including alkaloids, flavonoids, coumarins, limonoids, terpenoids, glycosides, and polyphenols. These constituents were confirmed through retention times and MS/MS spectra against commercial reference standards, natural product databases, and research findings.

Oral administration of VYAC at 150 and 300 mg/kg for 21 days substantially improved lesions on the dorsal skin of mice, and in turn, reduced erythema, excoriation, and scaling. The clinical scores for AD symptoms were significantly lower in the VYAC-treated groups compared to the AD model group. Additionally, histology showed that VYAC treatment reduced epidermal and dermal thickness, as well as mast cell infiltration, demonstrating its efficacy in repairing the skin barrier and mitigating inflammatory responses. Furthermore, VYAC was shown to decrease serum levels of IgE and histamine, which were elevated in the AD model group, indicating its potential to moderate immune responses associated with AD. The study also highlighted VYAC’s ability to reduce mRNA levels of proinflammatory markers, Syk, IL-4, and TNF-α, and suppress NF-κB expression, suggesting a significant anti-inflammatory mechanism. In vitro experiments with RBL-2H3 cells indicated that VYAC inhibited mast cell degranulation and cytokine secretion, further supporting its role in managing anaphylactic and inflammatory processes. By blocking the phosphorylation of Syk and the NF-κB pathway, VYAC demonstrates a reduction in key signaling events that contribute to AD pathogenesis triggered by mast cells [54].

## 4. Challenges and Future Directions

Despite the promising findings, several challenges remain in the development and clinical translation of natural compound-based therapies for allergic diseases. These include issues related to bioavailability, standardization of extracts, the sustainability and sourcing of raw materials, and potential adverse effects. Furthermore, the complexity of allergic reactions and individual variability in response to treatment necessitate further research to elucidate optimal dosing regimens and identify predictive biomarkers of treatment response. Future studies should also explore the potential synergistic effects of combining small-molecule natural compounds with existing therapies (Figure 4), as well as investigate novel delivery strategies to enhance their efficacy and safety. The sourcing of high-quality medicinal herbs is extremely important and can be a rate-limiting factor in the wide adoption of natural compound-based therapies. But above that, the typically high dose of herbal formulations can be a hurdle for adherence to treatment, so isolating and concentrating the highest-efficacy small molecules from herbal formulas can decrease the burden of a high pill load. Being able to isolate and synthesize bioactive compounds from plants in a lab can help to preserve medicinal plant populations in nature and allow for experiments that can be focused on a few high-efficacy molecules to better understand the specifics of their mechanisms of action on pathologic mast cells and basophils.

Traditional medicine has shown remarkable success in addressing complex diseases by modulating multiple pathways to restore malfunctioning organ systems to their normal state. However, understanding its mechanisms of action is challenging due to the presence of numerous complex compounds and their metabolites. Considering these treatments aim to target and suppress key effector cells in the immune system, it is important to note that many of these compounds have also been studied in an anti-cancer context and not in a cancer-causing context.

By combining the identification of new disease targets with the isolation of active compounds, the system pharmacology approach, along with high-throughput computational analysis, offers a potent means to investigate how traditional medicine works and the effects of small-molecule natural compounds in treating pathologic mast cells and basophils. A growing body of evidence utilizing system pharmacology provides a holistic understanding of drug effects on the human interactome, aiding in drug discovery, analyzing multiple agents for complex diseases, and predicting adverse reactions while also building pharmacology libraries of phytochemicals (Figure 5). In one such study, Wang et al. [70] laid a solid foundation for the systemic characterization of small-molecule natural compounds in traditional Chinese medicine (TCM) by identifying compound–target interactions and evaluating the potential therapeutic targets of traditional formulas like Shi Zhen Tea (SZT) in treating eczema [71]. This study helped in identifying compound–target interactions and evaluating the potential therapeutic targets of SZT, a formula composed of four Chinese herbs: *Sophorae flavescentis radix*, *Schizonepetae herba*, *Arctii fructus*, and *Atractylodis macrocephalae rhizoma*. The identification of active compounds and their interactions with specific targets provide a solid springboard for validation through in vitro and in vivo experiments. It is crucial to consider the broader implications of this research in the context of identifying other TCM formulas and a deeper exploration of the key pathways in which they are involved. This could pave the way for the development of new drugs or combination therapies in treating allergic inflammation.

Similarly, to fully understand the role of pathological mast cells and basophils in allergic diseases, it is also important to explore their interactome. The interactome refers to the network of all interactions within a cell or organism, such as signaling pathways and regulatory networks [72]. Allergic diseases like food allergy, asthma, rhinitis, and atopic dermatitis are complex, involving both allergen-specific IgE and non-IgE mechanisms [73]. Additionally, these diseases are shaped by genetic, epigenetic, and environmental factors that contribute to their diverse phenotypic presentations [74]. However, modifying any single cellular pathway can result in unintended downstream effects. This emphasizes the need for a comprehensive understanding of these interconnected pathways. In this context, the development of protein–protein interaction (PPI)-based network models has emerged as an approach to characterize the heterogenicity observed in allergic disease phenotypes. This allows for a bioinformatic-driven approach to the complex pharmacology inherent to TCM [75]. By integrating PPI-based network models with information from the traditional Chinese medicine system pharmacology database (TCMSP) [76,77] and emerging “-omics” techniques, such as mass spectrometry, spatial transcriptomics, and single-cell RNA sequencing, new insights into the relationships among herbs, ingredients, and targets can be gained. This approach supports the integration of traditional and modern medicine for drug discovery.

## 5. Conclusions

Allergic reactions depend on the activation of mast cells and basophils, which release inflammatory mediators upon allergen exposure. Despite their minority status among immune cells, the role of mast cells and basophils in allergy pathogenesis is pivotal, prompting the search for more effective treatments. Small-molecule natural compounds from botanical sources have emerged as promising candidates due to their diverse pharmacological properties and relatively low toxicity. Studies highlight compounds like berberine, sauchinone, and arctigenin, along with formulas such as FAHF-2 and SHL, as potential modulators of mast cell and basophil activity, offering hope for improved allergy management.

However, challenges persist in the development and clinical application of natural compound-based therapies. Issues like bioavailability, proper dosing, synergistic or antagonist effects, and long-term health need to be addressed. Future research directions should aim to address these challenges by integrating system pharmacology approaches and computational analysis to offer a robust framework for understanding the mechanisms of action of small-molecule natural compounds. By systematically characterizing compound target interactions and evaluating therapeutic targets, this approach facilitates drug discovery and the development of new treatments for allergic inflammation. Overall, the exploration of small-molecule natural compounds from botanical sources represents a promising frontier in allergy management, with the potential to provide safer and more effective long-term solutions to treat pathologic mast cells and basophils.

## Figures and Tables

**Figure 1 cells-13-01994-f001:**
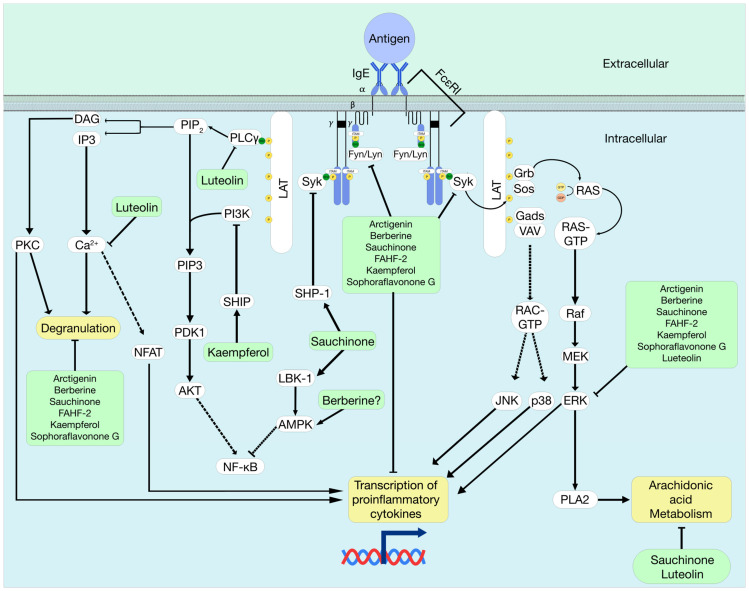
Proposed model of small-molecule interaction in the signaling pathways of FcεRI-mediated mast cell and basophil activation: The interaction between receptor-bound IgE and an antigen induces the crosslinking of multiple FcεRI receptors and initiates a network of signal transduction pathways that lead to the activation of mast cells and basophils. Upon the crosslinking of FcεRI, Lyn, and Fyn, phosphorylate ITAMs which recruit Syk and initiate downstream signaling pathways, including the PLCγ, PI3K, and MAPK pathways, culminating in degranulation and transcription. Negative regulators like SHP-1, SHIP, LKB1, and AMPK counterbalance these pathways, suggesting potential therapeutic targets for allergic diseases. Solid arrows indicate direct interaction, and dashed arrows indicate indirect interaction.

**Figure 2 cells-13-01994-f002:**
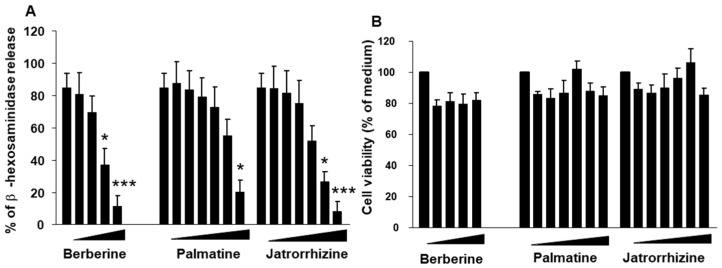
Major alkaloid compounds in fraction 2 from FAHF-2 by HPLC–inhibited mast cell degranulation in RBL-2H3 cells: (**A**) dose-dependent responses to berberine (1.25, 2.5, 5, and 10 µg/mL), palmatine (1.25, 2.5, 5, 10, and 40 µg/mL), and jatrorrhizine (1.25, 2.5, 5, 10, and 40 µg/mL) by RBL-2H3 cells; (**B**) cell viability: berberine, palmatine, and jatrorrhizine toxic effects on RBL-2H3 cells using MTT assay. * *p* < 0.05, *** *p* < 0.001 versus untreated control. Adapted with permission from ref. [33].

**Figure 3 cells-13-01994-f003:**
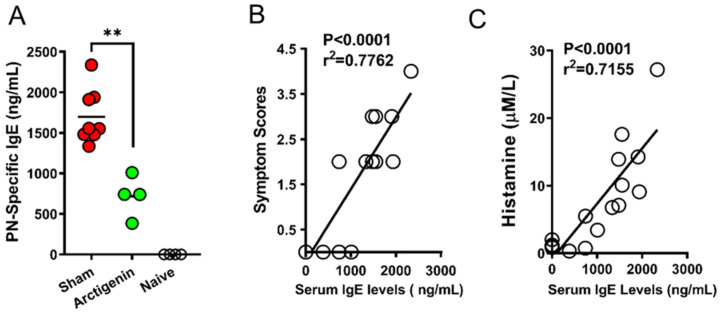
Effects of arctigenin on peanut-specific antibodies and protection of anaphylactic in a peanut-allergic mouse murine model. Serum peanut-specific IgE (**A**) and serum histamine (**B**) levels were measured by ELISA. ns means no statistical difference, ** *p* < 0.01 (*n* = 4–7). Correlations between PN-IgE and histamine levels (**C**) were evaluated. Each circle represents an individual mouse. Adapted with permission from ref. [40].

**Figure 4 cells-13-01994-f004:**
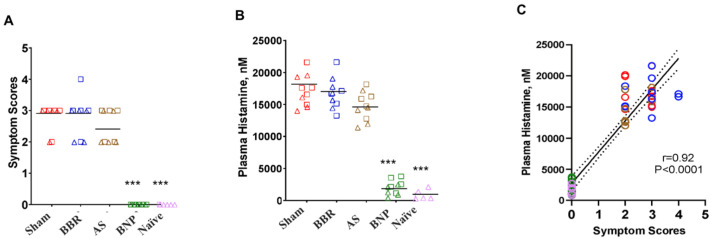
Long-term protection of berberine treatment with a boiled peanut oral immunotherapy at W30 and W70: (**A**) Symptom scores were assigned 30 min after the challenge using the following criteria: 0—no symptoms; 1—scratching and rubbing around the snout and head; 2—puffiness around the eyes and snout, pilar erecti, reduced activity, and/or decreased activity with increased respiratory rate; 3—wheezing, labored respiration, and cyanosis around the mouth and the tail; 4—no activity after prodding, or tremor and convulsion; 5—death. (**B**) Plasma histamine levels were measured by duplicate ELISA of individual plasma samples harvested from blood collected 30 min after the measurement of body temperatures. (**C**) Analysis of correlation between symptom scores and plasma histamine at W30 and W70 challenge time points. Color key for symbols: red—sham, blue—BBR, brown—AS, green—BNP, pink—naïve. Bars in (**A**) are group medians and in (**B**) are group means. In (**C**), the solid line represents the regression line, and the dashed line represents the 95% confidence interval. Red color represents sham, blue color represents BBR, tan color represents AS, green color represents BNP, and purple color represents naïve. N = 5 mice/group. Data represent 10 readouts from a combination of W30 (square symbols) and W70 (triangle symbols) challenges. *** *p* < 0.001 vs. sham. BBR: berberine; AS: *Angelica sinensis;* BNP: berberine-containing natural medicine with a boiled peanut oral immunotherapy. Adapted with permission from ref. [70].

**Figure 5 cells-13-01994-f005:**
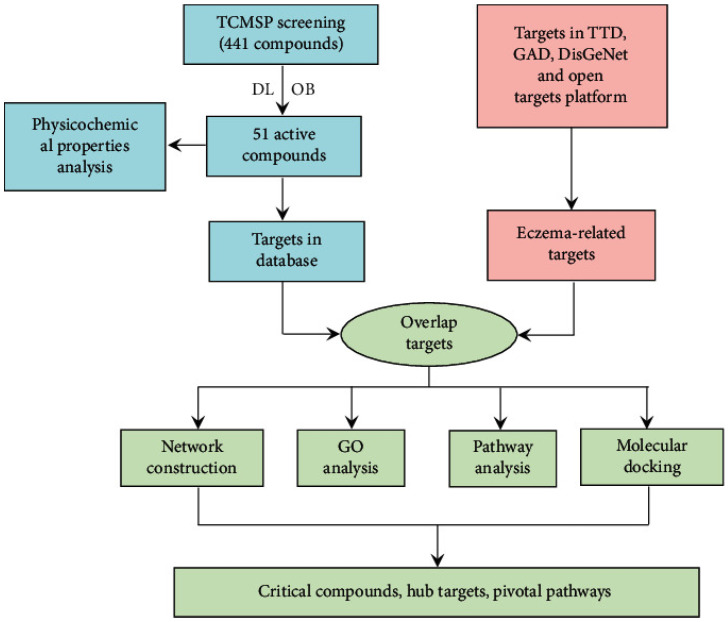
The workflow of systems pharmacology used for the analysis of the SZT formula for eczema treatment. TCMSP: TCM system pharmacology; OB: oral bioavailability; DL: drug-likeness; TTD: therapeutic target database; GAD: genetic association database; and GO: gene ontology. First, 51 active compounds of formula were selected from the TCMSP database. Their biological targets were further collected from several reliable databases. Meanwhile, physicochemical property analysis was conducted to summarize common properties of active compounds in different physicochemical parameters. Similarly, the targets involved in the pathological process of eczema were collected from related databases, including TTD, GAD, DisGeNet, and Open Targets Platform. With biological targets of compounds and related targets of eczema, their shared targets were chosen as potential therapeutic targets of SZT for eczema. Then, network construction, GO analysis, pathway analysis, and molecular docking were used to reveal the critical compounds from the SZT, the hub targets, and the pivotal regulated pathway of SZT for eczema treatment. Adapted with permission from ref. [71].

**Table 1 cells-13-01994-t001:** Selected small-molecule natural compounds and their mechanisms of action. Up arrows indicate increase and down arrows indicate decrease.

Compound	Botanical Source	Mechanism of Action	Evaluated Model	Refs.
Berberine	*Coptis chinensis* *Phellodendron japonicum* *Berberis aquifolium*	↓ Lyn, Syk, Gab2 phosphorylation ↓ β-hexosaminidase and histamine release ↓ Phosphorylated JNK, ERK, p38 ↓ IL-4 and TNF-α production ↓ Serum IgE ↓ Clinical dermatitis score severity and spontaneous scratching behavior in mice ↓ Degranulated mast cells ↓ mRNA levels of CCL11, MIF, IL-4, IL-5 ↓ Protein expression of ELF3F, MALT1	RBL-2H3 cells, Sprague Dawley rats, rat peritoneal mast cells, BALB/c mice, C3H/HeJ mice, MC/9 cells, human skin-derived mast cells, mouse peripheral blood leukocytes	[18,33,34,35,36,37]
Sauchinone	*Saururus chinensis*	↓ LTC4 and PGD2 levels ↓ PCA reaction through Evans blue dye extravasation ↑ Phosphorylated LKB1, AMPK, and ACC ↓ PLCγ1, ERK1/2, JNK, and IKK activation ↑ SHP-1 phosphorylation	AMPKα2−/− mice, BALB/cJ mice, C57BL/6 mice, mouse BMMCs,	[38,39]
Arctigenin	*Arctium lappa* *Asteraceae* family	↓ Allergen-specific IgE production and Th2 cytokines (IL-5, IL-13) ↓ Plasma histamine levels ↓ Proinflammatory cytokines (IL-1β, IL-6, IL-8, TNF-α) ↓ Proinflammatory chemokines (CCL2, CCL3, CCL4, CCL5) ↓ Phosphorylated Lyn, Fyn, and Syk ↓ Phosphorylated Akt PLCγ1/2 and PKCδ ↓ Phosphorylated, ERK, JNK, and p38 ↓ Nuclear translocation of NF-κB ↓ RIP2/caspase-1 activation	IgE-producing human myeloma U266 cells, C3H/HeJ mice, PBMCs from food-allergic patients, HMC-1 cells, RBL-2H3 cells, ICR mice	[40,41,42]
Sophoraflavanone G	*Sophora flavescens*	↓ Syk phosphorylation ↓ β-hexosaminidase release ↓ Proinflammatory cytokines IL-4 and TNF-α ↓ Phosphorylation of PLCγ1, AKT, p38, ERK1/2, JNK ↓ PCA response ↓ Protein levels of COX-2 and IL-4	3D pharmacophore model and molecular docking screen, RBL-2H3	[20]
Kaempferol	*Ginkgo biloba* *Rosmarinus officinalis*	↓ PCA response ↓ Degranulated mast cells through histological analysis ↓ Serum histamine levels ↑ Body temperature during allergen challenge ↓ β-hexosaminidase release ↓ TNF-α, IL-8, and CCL2 protein levels ↓ Intracellular Ca^2+^ influx ↓ Phosphorylated PLC, IP3R, PKC, AKT, NF-κB, Lyn, Syk, and DJ-1 ↓ Surface expression of FcεRI ↑ mRNA and protein levels of SHIP-1	LAD2 cells, C57BL/6 mice, BMMCs	[43,44]
Luteolin	*Reseda luteola*	↓ Serum levels of β-hexosaminidase release, 5-HT, histamine, tryptase, MCP-1, PGD2, TNF-α, IL-8, and IL-13 ↓ Intracellular Ca^2+^ influx ↓ Lyn, Btk, PLC-γ activation ↓ Release of LT and GM-CSF ↓ Phosphorylation of PKC, ERK, JNK	LAD2 cells, C57BL/6 mice, human cultured mast cells	[45,46]
*Shuang Huang Lian* tea	*Lonicerae Japonicae Flos* *Fructus Forsythiae* *Scutellariae Radix*	↓ β-hexosaminidase release ↓ Cytosolic Ca^2+^ ↑ MCU activation	LAD2 cells, P815 cell, mouse	[47]
FAHF-2/EBF-2	*Prunus mume* *Zanthoxylum schinifolium* *Angelica sinensis* *Zingiber officinalis* *Cinnamomum cassia* *Phellodendron chinense* *Coptis chinensis* *Panax ginseng* *Ganoderma lucidum*	↓ Syk phosphorylation ↓ Mast cell and basophil population numbers ↓ Expression of FcεRI on mast cells ↓ Allergen-specific IgE	C3H/HeJ mice, MC/9 cells, RBL-2H cells, human skin mast cells, mouse peripheral blood leukocytes, human PBMCs, U266 cells	[33,48,49,50,51]
Xin-Yi-Qing-Fei-Tang	*Gypsum Fibrosum* *Bulbus Lilii* *Anemarrhena asphodeloides* *Eriobotrya japonica* *Ophiopogon japonicus* *Gardenia jasminoides* *Scutellaria baicalensis* *Magnolia biondii* *Glycyrrhizauralensis Fisher* *Cimicifuga heracleifolia*	↓ Airway hypersensitivity ↑ Respiratory system resistance (Rrs) and elastance (Ers) ↓ Serum IgE levels ↓ BALF inflammatory cell infiltration ↓ mRNA IL-3, IL-4, IL-13, TNF-α, GM-CSF, COX-2, ALOX-5, CCL2	BALB/c, RBL-2H3 cells	[52]
Jiu-Wei-Yong-An	*Smilax glabra* *Dioscorea oppositifolia* *Forsythia suspensa* *Dioscorea collettii* *Rehmannia glutinosa* *Plantago asiática* *Isatis tinctoria* *Alisma plantago-aquatica* *Angelica sinensis*	↓ Dermatitis severity, epidermal thickness, and mast cell aggregation ↓ Pruritus ↑ Skin moisture content ↓ JAK1/STAT3 signaling ↓ p38, ERK, and JNK, phosphorylation ↓ Production of TNF-α, IL-1β, IL-4, IL-13, IL-31, IL-33, and IFN-γ	network pharmacology, molecular docking, BALB/c mice	[53]
*Viola yedoensis* Makino anti-itching compound	*Viola yedoensis* *Sophora flavescens* *Dictamnus dasycarpus*	↓ Skin lesions ↓ Epidermal and dermal thickness ↓ Mast cell infiltration ↓ Serum levels of IgE and histamine ↓ mRNA levels of Syk, IL-4, TNF-α ↓ Syk phosphorylation and NF-κB phosphorylation ↓ β-hexosaminidase release	BALB/c mice, RBL-2H3 cells	[54]

## Data Availability

No new data were created or analyzed in this study. Data sharing is not applicable to this article.

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
