# Peer review of "Effect of Small-Molecule Natural Compounds on Pathologic Mast Cell/Basophil Activation in Allergic Diseases"

_cells, 2024, doi:10.3390/cells13231994_

Round 1
Reviewer 1 Report
Comments and Suggestions for Authors
Allergic disorders are caused by inflammatory mediators such as leukotrienes, which are released from mast cells and basophils following the encounter between immunoglobulin E (IgE) proteins and Fc epsilon RI receptors on these cells. As a result, small molecules that interfere with this interaction could be effective in treating allergic disorders. In this context, some research is focused on the study of a series of small molecules of natural origin. By interfering with the interactions of essential proteins, these molecules show promising potential for antiallergic treatment. This review reports data from the literature on the effects of natural small molecule compounds derived from botanical sources on pathological mast cells and basophils, highlighting their potential in the management of allergies.
The topic reported in this review, concerning the search for effective treatments to improve the quality of life of millions of people affected by allergic diseases, is of great interest.
The introduction provides the reader with background information on the topic discussed.
The following sections are adequately divided and report sufficiently up-to-date literature references.
The conclusions report the aspects that need to be further investigated.
Overall, the review is well structured. However, some changes are required as indicated below.
The keywords are intended to facilitate web searches for the topic discussed. Therefore, please replace the terms already present in the title with different terms.
Authors should add a paragraph in the text on the meaning of interactome, i.e. the integrated network of all physical interactions within the cell, in order to clarify the relationship between molecular perturbations and phenotypic manifestations of diseases.
Comments on the Quality of English LanguageThe English language used in the manuscript is acceptable.
Author Response
comment 1: The keywords are intended to facilitate web searches for the topic discussed. Therefore, please replace the terms already present in the title with different terms.
Response 1: Thank you for pointing this out. I agree with this comment in that optimizing keywords will benefit the searchability of this article. I believe that the title of my review article is concise. However, I did change the keywords to facilitate web search optimization from Mast cell/basophil; Allergy; FcεRI; IgE; natural compounds to Mast cell/basophil activation; allergic diseases; pathologic IgE; small molecule natural compounds. Lines 26-2
comment 2: Authors should add a paragraph in the text on the meaning of interactome, i.e. the integrated network of all physical interactions within the cell, in order to clarify the relationship between molecular perturbations and phenotypic manifestations of diseases.
Response 2: I also agree with this comment. Thank you for pointing it out. I vaguely mention the idea of the interactome throughout the paper without actually naming it. I added a paragraph about the interactome from lines 648-664
Reviewer 2 Report
Comments and Suggestions for Authors
This well-written review article, it examines the functions of mast cells and basophils in allergic responses and explores the possibility of natural substances to limit their activation and degranulation. This review explores small molecule natural substances derived from plants, emphasising their processes and possible therapeutic applications in allergy control. However, it is necessary to address these concerns to enhance the review's comprehensiveness, impact, and accessibility:
1. The review highlights issues with bioavailability, although it might explore alternative remedies more thoroughly. Examine methodologies such as nanoparticle delivery, liposome encapsulation, and structural alterations to enhance bioavailability. Incorporating the newest breakthroughs in medicine delivery for natural chemicals will enhance value.
2. The review article emphasizes on allergic responses related to mast cells and basophils but does not thoroughly examine the potential impact of these natural substances on a wider spectrum of allergic illnesses, such as food allergies compared to respiratory allergies. Enhanced separation among diverse allergy diseases might provide a more focused comprehension of their prospective uses.
3. The possible synergistic effects of integrating these natural substances with traditional allergy therapies are very briefly addressed. Further investigation into the synergistic effects of these substances in conjunction with established treatments, such as monoclonal antibodies or immunotherapy, would provide a more comprehensive understanding of their complete potential.
Comments on the Quality of English LanguageThe grammar and English in the document are generally clear, but there are a still following areas where improvements can be made for better clarity and flow.
1. Some sentences are quite long and complex, making them harder to follow. Breaking these into shorter, more concise sentences can improve readability.
2. Be consistent with terminology. For instance, switching between terms like "natural compounds," "small molecule natural compounds," and "botanical compounds" might confuse readers. Choose one term and stick with it unless a distinction needs to be made.
3. The document uses the passive voice in many places. While this is acceptable in scientific writing, using more active voice where possible can make the text more engaging.
Author Response
comment 1: The review highlights issues with bioavailability, although it might explore alternative remedies more thoroughly. Examine methodologies such as nanoparticle delivery, liposome encapsulation, and structural alterations to enhance bioavailability. Incorporating the newest breakthroughs in medicine delivery for natural chemicals will enhance value.
Response 1: Thank you very much for this comment. It seems like you are a true expert in the field. I added a section about nanoparticle delivery and adjuvant herbs to address this shortfall. I focused on berberine nanoparticles because this is the research I am most familiar with. The new information can be found between lines 188-216.
Comment 2: The review article emphasizes on allergic responses related to mast cells and basophils but does not thoroughly examine the potential impact of these natural substances on a wider spectrum of allergic illnesses, such as food allergies compared to respiratory allergies. Enhanced separation among diverse allergy diseases might provide a more focused comprehension of their prospective uses.
Response 2: I completely agree with your comment. However, I aimed to concentrate specifically on pathologic mast cells and basophils in this paper rather than covering the entire spectrum of allergic diseases. While many allergic conditions are introduced, the complexity of these diseases is also discussed. I hope this addresses your concern. I plan to write another review article that will directly tackle this topic in greater depth.
comment 3: The possible synergistic effects of integrating these natural substances with traditional allergy therapies are very briefly addressed. Further investigation into the synergistic effects of these substances in conjunction with established treatments, such as monoclonal antibodies or immunotherapy, would provide a more comprehensive understanding of their complete potential.
response 3: Thank you for this comment. Although, I briefly mention immunotherapy and monoclonal antibodies in the intro and in small increments throughout the paper. I added a paragraph on the synergistic effects of a specific herbal formula (EB-2) used in combination with oral immunotherapy. As more research comes out I hope to update this further. lines 425-445
Response about clarity and flow: I completely agree. I wish I were a better writer. I fixed a few spots where I thought the language was confusing. I also changed all the terms to small molecule natural compounds. Thank you for the suggestion. I believe that this makes the paper stronger.